# Remote Sensing Big Data Classification with High Performance Distributed Deep Learning

**Rocco Sedona** [1,2,3,4,*,†] , **Gabriele Cavallaro** [2,3,4,†] , **Jenia Jitsev** [2,4,†] , **Alexandre Strube** [2] , **Morris Riedel** [1,2,3,4] and **Jón Atli Benediktsson** [1]

1    School of Engineering and Natural Sciences, University of Iceland, Dunhagi 5, 107 Reykjavík, Iceland; morris@hi.is (M.R.); benedikt@hi.is (J.A.B.)
2    Jülich Supercomputing Centre (JSC), Forschungszentrum Jülich (FZJ), Wilhelm-Johnen-Strasse 1, 52425 Jülich, Germany; g.cavallaro@fz-juelich.de (G.C.); j.jitsev@fz-juelich.de (J.J.); a.strube@fz-juelich.de (A.S.)
3    High Productivity Data Processing Research Group, JSC, 52425 Jülich, Germany
4    Cross-Sectional Team Deep Learning (CST-DL), JSC, 52425 Jülich, Germany
*    Correspondence: r.sedona@fz-juelich.de; Tel.: +49-2461-61-1497
†    These authors contributed equally to this work.

**Abstract:** High-Performance Computing (HPC) has recently been attracting more attention in remote sensing applications due to the challenges posed by the increased amount of open data that are produced daily by Earth Observation (EO) programs. The unique parallel computing environments and programming techniques that are integrated in High-Performance Computing (HPC) systems are able to solve large-scale problems such as the training of classification algorithms with large amounts of Remote Sensing (RS) data. This paper shows that the training of state-of-the-art deep Convolutional Neural Networks (CNNs) can be efficiently performed in distributed fashion using parallel implementation techniques on HPC machines containing a large number of Graphics Processing Units (GPUs). The experimental results confirm that distributed training can drastically reduce the amount of time needed to perform full training, resulting in near linear scaling without loss of test accuracy.

**Keywords:** distributed deep learning; high performance computing; residual neural network; convolutional neural network; classification; sentinel-2

## 1. Introduction

Modern Earth Observation (EO) programs have an open data policy and provide a massive volume of free multisensor data every day. Their systems have substantially advanced in recent decades due to the technological evolution integrated into Remote Sensing (RS) optical and microwave instruments [1]. NASA's Landsat [2] (i.e., the longest running EO program) and ESA's Copernicus [3] provide data with high spectral–spatial coverage at high revisiting time, which enables global monitoring of the Earth in a near real-time manner. Copernicus, with its fleet of Sentinel satellites, is now the World's largest single EO program (https://sentinel.esa.int/web/sentinel/missions). These programs are showing that the vast amount of raw data available call for re-definition of the challenges within the entire RS life cycle (i.e., data acquisition, processing, and application phases). It is not by coincidence that RS data are now described under the big data terminology, with characteristics such as volume (increasing scale of acquired/archived data), velocity (rapidly growing data generation rate and real-time processing needs), variety (data acquired from multiple satellites' sensors that have different spectral, spatial, temporal, and radiometric resolutions), veracity (data uncertainty/accuracy),

and value (extracted information) [4,5]. The Sentinel-2 mission, for instance, has been operating since June 2017 with a constellation of two polar orbiting satellite platforms, which allow a temporal resolution of 5 days at the equator (and even less for areas covered by more than one orbit). Both Sentinel-2A and Sentinel-2B are equipped with a Multispectral (MS) instrument which acquires 13 optical narrow bands in moderate-to-high spatial resolution (10, 20, and 60 m) and generates 23 TB/day of MS data. The freely available imagery from Sentinel-2 received major attention within the research community. From 1 December 2017 to 30 November 2018, the Sentinel Data Access System had a publication rate of over 26,500 products/day with an average daily download volume of 166 TB (https://sentinels.copernicus.eu/web/sentinel/news/-/article/2018-sentinel-data-access-annual-report). The large-scale, high-frequency monitoring of the Earth requires robust and scalable Machine Learning (ML) models trained over annotated (i.e., not raw) time series of multisensor images at global level [6,7] (e.g., acquired by Landsat 8 and Sentinel-2). However, these data do not exist yet. This is largely due to the inherent interpretation complexity of RS data (e.g., hyperspectral and RADAR data) and the effort and cost involved in the collection of training samples. This remains a key limiting factor in the RS community for the research and development of successfully operational Deep Learning (DL) classifiers for RS data.

Nevertheless, DL has already brought crucial achievements in solving RS image classification problems, working on raw multispectral satellite image data [8–10]. The state-of-the-art results have been achieved via deep networks with backbones based on convolutional transformations (e.g., Convolutional Neural Networks (CNNs) [11,12], Recurrent Neural Networks (RNNs) [13], and Generative Adversarial Networks (GANs) [14]). Their hierarchical architecture composed of stacked repetitive operations enables the extraction of useful image features from raw pixel data and modeling high-level semantic content of RS images. However, DL architectures have a much larger number of parameters to estimate than classic ML methods (e.g., shallow classifiers based on handcrafted features) [15]. Thus, their performance and generalization capabilities are considerably dependent on the amount and quality of available training data. That is, to train these networks, a very large annotated training set of sufficient diversity is needed in order to learn effective models.

Table 1 shows the main free annotated remote sensing datasets (i.e., for classification of RGB and MS images) that are currently available for benchmarking DL classifiers. The gap in terms of data size with the computer vision domain (e.g., ImageNet with 14,197,122 images (http://www.image-net.org/)) is still considerably high. Nonetheless, there is an evident trend towards datasets with a higher number of annotated samples and degree of classification complexity (e.g., BigEarthNet [16], a multiclass classification task of 590,326 images). Consequently, the computational intensity and memory demands of DL will continuously increase in the future. In this scenario, approaches relying on local workstation machines (i.e., using MATLAB, R, SAS, SNAP, and ENVI for data analysis and interpretation), can provide only limited capabilities. Despite modern commodity computers and laptops becoming more powerful in terms of multicore configurations and GPUs, the limitations with regard to computational power and memory are always an issue when it comes to fast training of large high-accuracy models from correspondingly large amounts of data. Therefore, the use of highly scalable and parallel distributed architectures (such as clusters [17], grids [18], or clouds [19]) is a necessary solution to train DL classifiers in a reasonable amount of time, which can then also provide users with a high-accuracy performance in the recognition tasks. High-Performance Computing (HPC) systems can reach a performance in the order of petaflops (i.e., $10^{15}$ floating point operations per second) and are already delivering unprecedented breakthroughs [20]. It is important to observe that ML and DL algorithms have transformed the workloads and workflows that run on these systems, especially when compared to classic HPC simulation problems. DL algorithms require higher memory and networking bandwidth throughput capabilities, as well as optimized software and libraries to deliver the required performance. On the one hand, DL can lead to more accurate classification results of land cover classes when networks are trained over large RS annotated datasets. On the other hand,

deep networks pose challenges in terms of training time. In fact, the use of a large datasets for training a DL model requires the availability of non-negligible time resources.

**Table 1.** Non-exhaustive list of open remote sensing datasets for image classification.

| Datasets | Image Type | Image Per Class | Scene Classes | Annotation Type | Total Images | Spatial Resolution (m) | Image Sizes | Year | Ref. |
|---|---|---|---|---|---|---|---|---|---|
| UC Merced | Aerial RGB | 100 | 21 | Single/Multi label | 2100 | 0.3 | 256 × 256 | 2010 | [21] |
| WHU-RS19 | Aerial RGB | ∼50 | 19 | Single label | 1005 | up to 0.5 | 600 × 600 | 2012 | [22] |
| RSSCN7 | Aerial RGB | 400 | 7 | Single label | 2800 | – | 400 × 400 | 2015 | [23] |
| SAT-6 | Aerial MS | – | 6 | Single label | 405,000 | 1 | 28 × 28 | 2015 | [24] |
| SIRI-WHU | Aerial RGB | 200 | 12 | Single label | 2400 | 2 | 200 × 200 | 2016 | [25] |
| RSC11 | Aerial RGB | 100 | 11 | Single label | 1323 | 0.2 | 512 × 512 | 2016 | [26] |
| Brazilian Coffee | Satellite MS | 1438 | 2 | Single label | 2876 | – | 64 × 64 | 2016 | [27] |
| RESISC45 | Aerial RGB | 700 | 45 | Single label | 31500 | 30 to 0.2 | 256 × 256 | 2016 | [28] |
| AID | Aerial RGB | ∼300 | 30 | Single label | 10,000 | 0.6 | 600 × 600 | 2016 | [29] |
| EuroSAT | Satellite MS | ∼2500 | 10 | Single label | 27,000 | 10 | 64 × 64 | 2017 | [30] |
| RSI-CB128 | Aerial RGB | ∼800 | 45 | Single label | 36,000 | 0.3 to 3 | 128 × 128 | 2017 | [6] |
| RSI-CB256 | Aerial RGB | ∼690 | 35 | Single label | 24,000 | 0.3 to 3 | 256 × 256 | 2017 | [6] |
| PatternNet | Aerial RGB | ∼800 | 38 | Single label | 30,400 | 0.062∼4.693 | 256 × 256 | 2017 | [31] |
| BigEarthNet | Satellite MS | 328 to 217,119 | 43 | Multi label | 590,326 | 10,20,60 | 120 × 120<br>60 × 60<br>20 × 20 | 2018 | [16] |

The objective of this contribution is to show that HPC systems speed up the training of DL networks through distributed training frameworks, which can exploit the parallel environment of HPC clusters. The distribution of the model among multiple nodes can considerably speed up the process of training it. This enables deployment of various models and comparison of their performances in a reasonable amount of time. The training of the model is performed via a multimachine data parallelism strategy that allows minimizing the time required to finish full training: The processing is distributed across multiple machines connected by a fast dedicated network (i.e., InfiniBand). This paper proposes a high-performance distributed implementation of the Residual Network (ResNet) [32] type of deep convolutional networks (so-called deep residual networks) for the multiclass RS image classification problem. The experiments are performed with the BigEarthNet [16] dataset over the HPC systems that are based at the Jülich Supercomputing Centre. The experimental results attest that distributed deep neural network training can extremely reduce the amount of time that is required to complete the training step without affecting prediction accuracy.

## 2. Deep Learning

### 2.1. The ResNet

ResNet is a deep residual CNN architecture developed by He et al. in [32] to overcome difficulties in training networks with a very large number of layers (>20, up to 1000 layers and more possible [33]), winning the ImageNet competition in 2015 [34]. The first instantiations of deep feed-forward CNNs were the ones providing groundbreaking advances in the field of computer vision on tasks like object detection and object recognition, outperforming previous state-of-the-art ML methods by large margins, e.g., AlexNet with 8 layers [35], VGG with 16 layers [36] or GoogleNet (Inception) with 22 layers [37]. An increasing number of processing layers resulted in further increasing accuracy performance on ImageNet challenges in terms of class recognition rates (the ImageNet-1k challenge has 1000 different object classes that have to be successfully learned during the training on 1.2 Million images [35,38]).

However, simply increasing the number of layers further by stacking more and more convolutional and other layers (pooling, etc) on top of each other was not functionally successful. The training of very deep networks resulted in worse accuracy, contrary to expectations set by previous results. It has been noted that degradation of the training accuracies may be partly caused by a phenomenon known as vanishing (or exploding) gradients. ResNet architecture has been designed to overcome this issue by introducing so-called residual blocks featuring skip connections. These connections implemented an explicit identity mapping for each successor layer in a deep network in addition to the learned operations that were applied to the input before it reaches the next layer [32,33]. The network was thus forced to learn residual mappings corresponding to useful transformations and feature extraction on the image input, while loss gradients could still flow undisturbed during the backward pass via available skip connections through the whole depth of the network. Different ResNet networks were shown to train successfully with a number of layers that was impossible to handle before, while using a smaller number of parameters than previous, less deep architectures (e.g, VGG or Inception networks), thus allowing for faster training.

ResNet-50 (where the number indicates the number of layers) has since then established a strong baseline in terms of accuracy, representing good trade-off between accuracy, depth, and number of parameters, in the same time being very suitable for parallelized, distributed training. As it still remains the strong baseline for object recognition tasks and is also widely used in scenarios for transfer learning ([39–41]), the ResNet-50 architecture is adopted for experiments to show successful distributed training for multiclass, multilabel classification from RS multispectral images.

### 2.2. Distributed Frameworks

Despite the permanently increasing computational power of Central Processing Unit (CPU)- and Graphics Processing Unit (GPU)-based hardware and essential improvements in efficiency of deep

neural network architectures like ResNet, it remains still a computationally very demanding procedure to train a particular deep neural network to successfully perform a challenging task like object recognition. Even with state-of-the-art hardware like NVIDIAs V100, full training of a ResNet-50 object recognition network on ImageNet-1k dataset of 1.2 Million images using a single GPU can still take more than one day on a single workstation machine (also when taking into account possible acceleration via more efficient mixed-precision (fp16 and fp32) training or special optimized computational graph compilers like TensorFlow's XLA). To conduct a multitude of experiments with various network architectures on large datasets, training therefore constitutes a prohibitively time-expensive procedure.

To overcome these limitations imposed by computationally expensive training, the DL community envisages different methods that enable distributed training across multiple computing nodes of clusters or HPC machines equipped with accelerators like GPUs or highly specialized TPUs [42,43]. Using these methods, it became possible to perform distributed training of large network models without loss of task performance and drastically reduce the amount of time necessary for a complete training. For instance, the time to fully train an object recognition network model on ImageNet-1k (1.2 Millions of images, ca. 80–100 epochs necessary for training to converge) was reduced by orders of magnitude only within a few years from almost one day to few minutes without substantial loss in recognition accuracy [44,45].

This work relies on a certain type of distributed training to conduct scaling experiments and make use of Horovod—a software library that offers a convenient way to execute training and supports TensorFlow and Keras [46]. Using Horovod, only a few modifications in the standard code used for quick single node model prototyping are necessary to adapt it for distributed execution across many nodes.

To enable distributed training, Horovod adapts a data parallel scheme. In the data parallel scheme, it is assumed that a network model to be trained can fit into the memory of a single GPU device. Many so-called workers can be then instantiated during the training, each occupying one available GPU. Each worker contains a clone of the network to train and gets a separate portion of data to train on, so that for each model update iteration, the global data mini-batch is split into different portions that are assigned to each worker. Working on their own portion of the mini-batch, each worker performs a forward pass to compute the network activations and the local loss given their current input, and a backward pass to compute the local gradients.

To keep all the network models across workers in sync, Horovod employs a decentralized, synchronous update strategy based on Ring-AllReduce operations [46,47], where gradients of all workers are collected, averaged, and applied to every clone model network to update their parameter weights. This is in contrast to centralized update strategies that usually require so-called parameter servers (PS) to communicate model parameters to the workers.

However, those implementations rely on TCP/IP internode communication, which is not available on our machines. On the other hand, Horovod relies on operations based on MPI and NCCL libraries, thus being our preferred choice.

The decentralized update makes better use of network topologies connecting the respective machines and thus usually employs a more efficient, homogeneous communication strategy to perform distributed training. On the one hand, the centralized parameter server-based update strategy offers the flexibility to add or remove the workers, which requires only reconfiguration of a parameter server. On the other hand, the decentralized approach may offer higher fault tolerance in terms of not having one weak spot in the communication chain—when a parameter server fails, it is hard to resume training; when a worker node fails, communication in the decentralized approach can still be reconfigured without affecting training, as every other working node possesses a full copy of the model.

For less reliable cluster systems, decentralized updates are therefore a viable option. For robust HPC systems, where note failure is rare, centralized schemes can be a performant choice as well. However, to avoid bottlenecks in communication during large-scale distributed training on HPC,

the setup of many PS is required, which complicates resource allocation, increases complexity of the necessary code, and makes proper training implementation difficult [42]. Thus, using a decentralized update scheme as employed by Horovod is an efficient choice in terms of simplicity and speed for distributed training on HPC.

As a high-level framework at the top of deep learning libraries, Horovod uses well-established MPI CUDA-aware routines and relies on the NCCL library [46,48] for efficient and robust implementation of communication between workers that makes the best out of the available network topology and bandwidth. The choice for Horovod as library for efficient distributed training is also motivated by the ease, clear structure, and transparency of the necessary code modifications. The corresponding strategy can be as well implemented in pure TensorFlow via the distributed strategies framework [49]; however, the effort to rewrite a single node prototype code is still considerably more when compared to modifications required by Horovod. Horovod also supports a unified scheme for using it with other libraries (PyTorch, MxNet), which again minimizes the effort to deal with specific details of each respective framework when implementing distributed training.

Apart from issues regarding efficient communication of information necessary for model updates during distributed training across multiple nodes, there is a further aspect to be dealt with in the algorithmic challenge to perform distributed training. This aspect is rooted in the nature of the optimization procedure that performs actual loss minimization. The majority of the optimization methods used to minimize loss during training are different variations of Stochastic Gradient Descent (SGD). If training has to be distributed across a substantial amount of workers, the effective size of the global mini-batch has to grow. Optimization thus has to cope with mini-batch sizes that are substantially larger that those used for training on a single node. Large mini-batches (for ImageNet, in the order of a few thousand images per batch as compared to the standard mini-batch size of a few hundreds for single node training) lead to substantial degradation of performance, e.g., recognition accuracy, if used without any additional countermeasures [50]. This may be partly due to the very nature of SGD, which requires a certain amount of noise produced by the rather small sizes of mini-batches used for update steps.

Currently, there are different solutions to secure the same performance level achieved on a single node with small mini-batch sizes despite the essential increase of the effective mini-batch size during distributed training. In the core of the simplest solutions is the tuning of the learning rate schedule that uses warm-up phases before the training, scales the learning rate with the number of distributed workers, and reduces the rate according to a fixed factor after a fixed number of epochs [6,44,50]. More sophisticated strategies to deal with very large batch sizes (for ImageNet, for instance, greater than $2^{13} = 8192$) use adaptive learning rates that are tuned dependent on network layer depth and the value of computed gradients and progress of training, such as that employed in LARS (Layer-wise Adaptive Rate Scaling)—an adaptive optimizer dedicated to large-scale distributed training setting [45,51].

## 3. Experimental Setup

### 3.1. Data

The training of the models was carried out using the list of patches provided by BigEarthNet (http://bigearth.net/). BigEarthNet is an archive consisting of 590,326 patches extracted from 125 Sentinel-2 tiles (Level 2A) acquired from June 2017 to May 2018 [16]. A number of labels is associated with each patch. The 43 labels originate from the CORINE Land Cover (CLS) inventory of 2018, available for 10 European countries. According to [16], the number of labels for each patch varies between 1 and 12, being in 95% of the cases at most 5. The patches have 12 spectral bands: (a) the 3 RGB bands and band 8 at 10 m resolution (120 × 120 pixels), (b) bands 5, 6, 7, 8a, 11, and 12 at 20 m resolution (60 × 60 pixels), and (c) band 1 and 9 at 60 m resolution (20 × 20 pixels). Band 10 has been excluded since it is used mainly for cirrus detection [52]. BigEarthNet also provides a list of the patches

with a significant amount of the area covered by snow or clouds, making it possible to exclude them from the analysis [53]. Figure 1 shows an example of the patches.

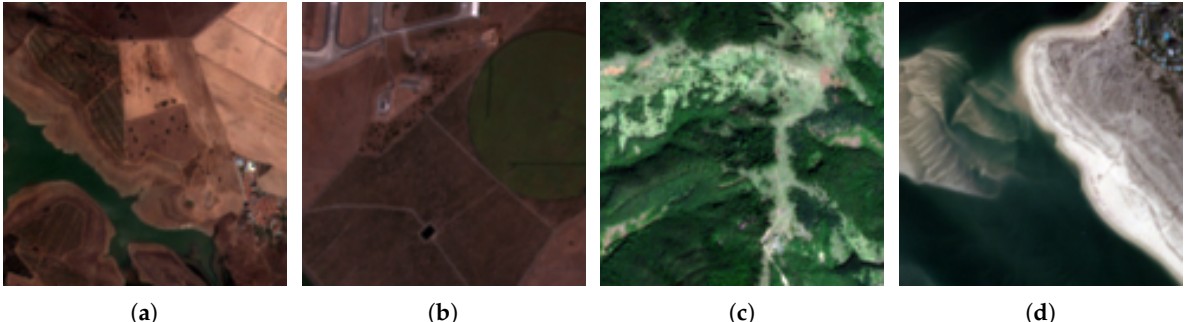

| (a) | (b) | (c) | (d) |

**Figure 1.** Example of patches: (**a**) agro-forestry areas, complex cultivation patterns, non-irrigated arable land, transitional woodland/shrub, water bodies, (**b**) airports, olive groves, permanently irrigated land, (**c**) broad-leaved forest, burnt areas, transitional woodland/shrub, (**d**) beaches/dunes/sands, estuaries, sea and ocean, and sport and leisure facilities.

### 3.2. Environment

The experiments were carried on two HPC sytems installed at the Jülich Supercomputing Centre: the Jülich Wizard for European Leadership Science (JUWELS) [54], and the Jülich Research on Exascale Cluster Architectures (JURECA) [55] supercomputers. In both machines, GPUs partitions were used: JUWELS consists of 46 nodes, with each having four NVIDIA V100 GPUs (with 16 GB of memory each), while JURECA has 75 nodes, each equipped with four NVIDIA K80 GPUs (with 24 GB of memory each). The available benchmark for the experiments relies on a maximum of 24 nodes (i.e., 96 GPUs) for each system.

For the evaluation, the following Python libraries were used: `TensorFlow 1.13.1`, `Keras 2.2.4`, `Horovod 0.16.2`, `Mpi4py 3.0.1` and `Scikit-learn 0.20.3`.

In order to upsample the Sentinel-2 bands at a lower resolution to the maximum resolution of 10 m, we use two different upscaling methods. One is based on the super-resolution deep network approach proposed by Lanaras et al. in [56]. Using super-resolved images, we can obtain the same high resolution across different bands. The authors provide a pretrained CNN model (i.e., `DSen2` (https://github.com/lanha/DSen2)) that was trained over a large Sentinel-2 training set which covers a wide range of geographical locations across different climate zones and land-cover types [56]. Another is based on simple standard bilinear interpolation. The simple upscaling is there to check whether there is any advantage in using an advanced super-resolution technique in our case.

The extraction of the patches was carried out with the Geospatial Data Abstraction Library `GDAL` `2.3.2` through its Python API. `GDAL` [57] is an open source programming library and set of utilities that facilitates the manipulation of raster data: It helps with data translation from different file formats, data types, and map projections.

### 3.3. Preprocessing Pipeline

One of the aims of this work is to evaluate models' performance that take Sentinel-2 patches as input, with all the multispectral bands upsampled to the resolution of 10 m for the RGB bands. The original BigEarthNet archive was used as a basis to extract the information for generating a new dataset, one that includes super-resolved patches, as well as the original ones (i.e., publicly available (http://hdl.handle.net/21.11125/921dbc5e-5948-4453-90c0-40b399ffa418)). In order to extract bands at a higher resolution, and to study whether those could help in enhancing the performances of the classification scheme, the DSen2 framework was employed to obtain patches in which the bands originally at a lower resolution (20 and 60 m) were super-resolved: In this way,

all bands become available at the maximum resolution of 10 m. DSen2 consists of two CNNs to perform the trained enhancement of the lower resolution bands into the highest resolution [56].

As shown in Figure 2, the first step in the preprocessing pipeline was to download the freely-available 125 Level 2A tiles from Copernicus Data Hub (https://scihub.copernicus.eu/). After that, the tiles were given as input to DSen2, and in this way, the upsampled tiles were computed. To extract BigEarthNet's original 519,226 patches with a low percentage of snow or cloud coverage, the approach described by the parallel Algorithm 1 was adopted. The algorithm computes the number of patches belonging to each Sentinel-2 tile and creates a matrix with the indices of the tile to be processed by each CPU, in such a way that the total amount of total patches is similar among all processors. With this strategy, idle time is avoided (e.g., a process that already extracted a small number of patches has to wait until other processes to finish their task). The algorithm was executed in parallel using 72 CPUs on JURECA.

The patches were saved in a single Hierarchical Data Format 5 (HDF5) [58] file. This format can be written and read in parallel. It has been organized by associating the data with different keys: "data_super" is the key of the datacube with the 12 upsampled multispectral bands, "data_10m", "data_20m" and "data_60m" stands for datacubes of bands at the original resolution of 10, 20, and 60 m. respectively, and "classes" includes the labels of each patch already binarized.

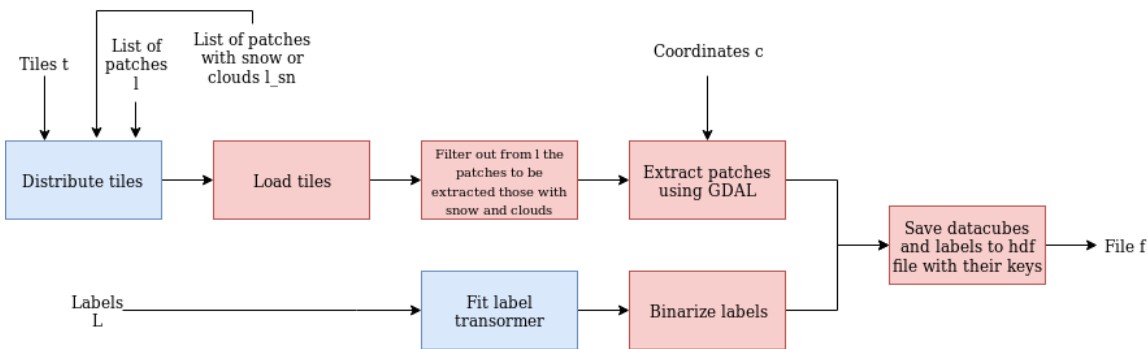

**Figure 2.** Preprocessing pipeline: extraction of the Sentinel-2 patches and their corresponding classes. Patches covered in snow and clouds are excluded.

---

**Algorithm 1** Distribution of tiles

---

Input: input parameters $n$ number of CPUs and $t$ tiles

Output: matrix $M$ with indices of tiles per processor

---

1: $M \leftarrow range(n_{proc})$
2: **if** $n - size(t) > 0$ **then**
3:     **for** $i \leftarrow 1$ to $len(t)/n$ **do**
4:         $arr \leftarrow zeros([n_{proc}])$
5:         $arr \leftarrow t.values[0 : n_{proc}]$
6:         $M \leftarrow vstack([M, flip(pad(range((i+1) \times n_{proc}, (i+1) \times n_{proc} + len(b), 1), (0, len(arr) - len(b)))$
7: **else**
8:     $M \leftarrow t$
9: **return** M

---

### 3.4. Multilabel Classification

Even though in some RS applications, the use of a single label per sample of a scene may be sufficient for a correct classification, there are cases where it might not be sufficient. As stated in [59],

an image of a beach could be correctly classified with a single label, without the need for separated labels such as "sand", "sea", or "buildings". Multilabel classification is defined as that type of classification where classes associated to each sample are not mutually exclusive [60].

According to [59], however, more complex scenarios require finer-grain labeling. For instance, distinguishing between images of urban areas with different building densities would require specific classes, which may also occur in combination with the presence of other classes, such as "road" or "green area". A characteristic of multilabeling classification is, in fact, that the occurrence of some class could be correlated with those of others appearing in similar scenarios.

The standard approach when it comes to the computation of the loss function in the multilabel classification case is the binary cross entropy. A vector of dimension equal to the number of classes is associated to each sample, where every vector cell represents the presence or absence of a specific class. In this way, the problem can be dealt with as a binary classification problem for each of the classes, hence treating them independently. The activation function used for multilabel classification is the sigmoid function, squashing all the elements of the label vector between 0 and 1. This is different from using the softmax activation function, which transforms the probabilities so that they sum up to 1. Instead, using the sigmoid function, it is possible to assure that the labels are not mutually exclusive in the multilabel case, but more than one can be associated to each sample.

### 3.5. Restricted RGB and Original Multispectral ResNet-50

Two configurations have been considered for the experiments to establish baselines for successful training. They differ according to the data in input: (a) input is limited to three RGB bands only, and (b) input contains 12 multispectral bands. The motivation is, on the one hand, to prepare grounds for transfer learning experiments using ImageNet pretraining on the data that contain RGB channels only. On the other hand, RGB configuration serves as a minimal baseline to check whether a full multispectral input can provide any additional boost for classification performance within standard ResNet architecture.

The classification scheme used in this paper is based on a slightly modified version of ResNet-50. In the present work, some changes to the model have been made to better adapt it to the land cover classification problem. The output layer has been modified to output the prediction probabilities for the 43 CLC classes. The input size has been changed from the original size of 224 × 224 pixels for each image to the size of the patches (i.e., 120 × 120 for the 10 m, 60 × 60 for the 20 m, and 20 × 20 for the 60 m resolution). Two different kinds of regularization have been adopted to reduce the risk of overfitting: (1) an L2 regularization has been applied to all convolutional layers to penalize large weights, and (2) a dropout with probability equal to 0.5 has been placed before the model's last dense layer.

Two data augmentation techniques were used. The first one is a simple rotation of 90, 180 or 270° and a flip operation, applied randomly to the patches.The second method is called a mix-up and consists in taking a batch and subtracting from it a shuffled version of itself, with a probability drawn from a beta distribution for each patch [61]. The use of these virtual augmented data created with a simple linear combination of the original samples encourages the model to learn smoother decision boundaries, making it more robust when unseen samples are fed into the network. An SGD with Nesterov momentum was selected as an optimizer [62]. The initial learning rate was computed as $\eta = 0.1 \frac{kn}{256}$ [50], where $k$ is the number of workers (i.e., GPUs) and $n$ is the batch size for each worker, which in this paper is set to 64. In our work, a step decay learning annealing schedule was used: The actual learning rate was computed multiplying by 0.1 the original learning rate after 30 epochs, by 0.01 after 60 epochs. and by 0.001 after 80 epochs. In our work, we trained the models for a total of 100 epochs. This technique is used to reduce the probability of the model to get stuck in a plateau using a too small learning rate, while on the other hand, a learning rate which is too high may cause an instability in the optimization process [63]. A warm-up of 5 epochs was applied at the start of the training process.

## 4. Results

### 4.1. Classification

The classification results are presented for the RGB and the multispectral models. Both models were adapted to the problem of multilabel classification from the original ResNet-50 [32]. For the performance metric of the experiment, we employed the F1 score, which is widely used for multiabel image classification problems. In Tables 2 and 3, the prediction results for a single experiment performed over 1 node of JUWELS (i.e., 4 NVIDIA V100 GPUs) are reported. For this proposed ResNet-50 architecture, the model trained on RGB bands performs almost as well as the multispectral model (see Table 2 that shows the global scores). The prediction scores of each individual class are reported by Table 3. It can be seen that some classes have a very high F1 score: e.g., the class "Sea and ocean" has a high F1 score. This is not surprising due to to the specific distinguishable spectral signature of water. For the same reason, the class "Coastal lagoons" is also easily detected by the model, despite heavy imbalance—this class has a much smaller number of samples compared, for instance, to "Sea and ocean".

**Table 2.** Classification results for the RGB and multispectral model: P precision, R recall and F1 score.

|  | P | R | F1 |
| --- | --- | --- | --- |
| RGB | 0.82 | 0.71 | 0.77 |
| multispectral | 0.83 | 0.75 | 0.79 |

**Table 3.** Classification results of each class for the RGB and multispectral model: F1 score and support for each class considering the test set.

|  | Support | F1 (Multispectral) | F1 (RGB) |
| --- | --- | --- | --- |
| Agro-forestry areas | 5611 | 0.803621 | 0.795872 |
| Airports | 157 | 0.300518 | 0.374384 |
| Annual crops associated with permanent crops | 1275 | 0.457738 | 0.442318 |
| Bare rock | 511 | 0.604819 | 0.620192 |
| Beaches, dunes, sands | 319 | 0.695810 | 0.608964 |
| Broad-leaved forest | 28,090 | 0.791465 | 0.771761 |
| Burnt areas | 66 | 0.029851 | 0 |
| Coastal lagoons | 287 | 0.884758 | 0.880294 |
| Complex cultivation patterns | 21,142 | 0.722448 | 0.698238 |
| Coniferous forest | 33,583 | 0.874152 | 0.866716 |
| Construction sites | 244 | 0.234482 | 0.213058 |
| Continuous urban fabric | 1975 | 0.784672 | 0.517737 |
| Discontinuous urban fabric | 13,338 | 0.780262 | 0.722825 |
| Dump sites | 181 | 0.287037 | 0.268518 |
| Estuaries | 197 | 0.699088 | 0.585034 |
| Fruit trees and berry plantations | 875 | 0.452648 | 0.417887 |
| Green urban areas | 338 | 0.387750 | 0.369477 |
| Industrial or commercial units | 2417 | 0.552506 | 0.556856 |
| Inland marshes | 1142 | 0.408505 | 0.364675 |
| Intertidal flats | 216 | 0.635097 | 0.584126 |
| Land principally occupied by agriculture | 26,447 | 0.686677 | 0.667633 |
| Mineral extraction sites | 835 | 0.507598 | 0.490980 |
| Mixed forest | 35,975 | 0.834221 | 0.797793 |
| Moors and heathland | 1060 | 0.561134 | 0.430953 |
| Natural grassland | 2273 | 0.569581 | 0.512231 |
| Non-irrigated arable land | 36,562 | 0.865387 | 0.839924 |
| Olive groves | 2372 | 0.621071 | 0.541914 |

**Table 3.** *Cont.*

|  | Support | F1 (Multispectral) | F1 (RGB) |
|---|---|---|---|
| Pastures | 20,770 | 0.780565 | 0.771802 |
| Peatbogs | 3411 | 0.535477 | 0.690319 |
| Permanently irrigated land | 2505 | 0.675662 | 0.643835 |
| Port areas | 93 | 0.503597 | 0.522388 |
| Rice fields | 709 | 0.669542 | 0.604770 |
| Road and rail networks and associated land | 671 | 0.300785 | 0.268623 |
| Salines | 75 | 0.608000 | 0.517857 |
| Salt marshes | 264 | 0.568578 | 0.532299 |
| Sclerophyllous vegetation | 2114 | 0.762123 | 0.671300 |
| Sea and ocean | 13,964 | 0.909013 | 0.979917 |
| Sparsely vegetated areas | 261 | 0.483460 | 0.380681 |
| Sport and leisure facilities | 996 | 0.367029 | 0.406827 |
| Transitional woodland/shrub | 29,671 | 0.664189 | 0.639412 |
| Vineyards | 1821 | 0.564012 | 0.545454 |
| Water bodies | 11,545 | 0.858107 | 0.823858 |
| Water courses | 1914 | 0.803948 | 0.737060 |

*4.2. Processing Time*

The processing times of the JURECA and JUWELS systems are reported only for the multispectral model. Due to the limited amount of computing time (i.e., core hours) allocated for this project, each experiment has been run only twice. Figures 3 and 4 report the mean and standard deviation values. It can be observed that the training time using two nodes (i.e., 8 GPUs) is half (172 s for an epoch on JUWELS) of the time required to execute the same training with one node (i.e., 4 GPUs) (347 s). The same can be said in the cases where 2 vs. 4 (172 s vs. 86 s), 4 vs. 8 (86 s vs. 42 s) and 8 vs. 16 (42 s vs. 20 s) nodes are considered. However, the scaling between 12 and 24 nodes seems to be less than linear (27 s vs. 15 s).

The use of this distribution approach has allowed us to reduce the total time for a full training on JUWELS from almost 35,000 s using 4 GPUs on 1 node to less than 2500 s using 96 GPUs on 24 nodes. The results on JURECA shown in Figure 4 confirm this observation. Although it can be seen that the full run on JURECA (on 2 nodes approximately 14 h, as can be seen in Figure 5) takes almost 3 times more time than those run on JUWELS (on 2 nodes in less than 5 h) due to the available GPUs (K80 vs. V100), on the other hand, taking advantage of this parallelization framework has enabled the full training of the model using older GPUs in a reasonable amount of time.

## 5. Discussion

The class imbalance poses a serious caveat on the performances of the models. In fact, it can be observed that there are classes which are heavily under-represented compared to others—e.g., in the test subset considered for this work, there are more than 30,000 patches associated with the label "Coniferous forest" but just 93 with label "Port areas". Thus, it comes as no surprise that the F1 score obtained for the classes with a low support (i.e., low number of samples) is on average less than the F1 score of the most populated classes of the dataset, since it is known that class imbalance can cause a bias towards the majority class [64]. As reported in Section 3.5 in this work, two simple data augmentation techniques were applied. However, the problem of class imbalance may require the use of of different techniques, e.g., upsampling of the under-represented samples [65] or loss weighting to let the model give more importance to samples associated with classes present in a lesser amount [64]. These methods should be implemented and tested in future work.W Another limitation that stems from the imbalance problem is that the spectral signature (i.e., the radiation reflected by the surface as a function of the wavelength) of areas associated with some classes could change over time, causing low classification results. That may be the case for the class "Burnt areas" (an example in Figure 1c), showing a very low F1 score. An approach to deal with such a class could be the adoption

of a multitemporal analysis, implementing, for instance, a change detection method to identify when a significant change in the spectral signature of a patch (such as the one caused by a fire) occurs. Moreover, CLS classes may be semantically too stringent for the purpose of classification of land cover using optical data alone. As an example, CLS has two different classes for "Discontinuous urban fabric" and "Green urban areas", which may represent patches with a similar information content. One last point which could be considered is the fact that the presence of some classes may be correlated with those of another one. For instance, it is reasonable to assume that "Beaches, dunes, sands" is correlated with the presence of classes associated with water, as can be observed in Figure 1d, or that a cultivation pattern is present at the same time of arable land as in in Figure 1a. In this work, it has not been used a method to explicitly take this information into account, as, e.g., it was done in [16], where the local descriptors generated by a CNN were then updated using an LSTM network on subtiles of the patches.

In Section 3.2, we stated that our work makes use of DSen2 to upsample the patches to the same resolution of 10 m across the different bands. We used DSen2 since it is a well-established method for super-resolution. However, an experiment in which we used a simple bilinear interpolation, run on 8 nodes on JUWELS, showed a very similar F1 score to those obtained using DSen2 (shown in Figure 6). Further studies should be conducted to investigate whether different DL models could take advantage of the enhanced spectral characteristics provided by DSen2.

Section 4.1 mentions that the model trained on RGB bands obtains a slightly lower average F1 score to the one achieved by the multispectral model. However, for the class airports, bare rock, peatbogs, port areas, sea ocean, and sport and leisure facilities, the F1 score of RGB is higher. For these classes, the model that is trained with the multispectral data is not able to isolate the RGB information from the other bands. Generally, a correct network architecture should deliver at least the same classification results (since multispectral data include the same RGB bands). As we explain in Section 2.1, we selected the ResNet-50 since it is a well-established baseline architecture in terms of accuracy, represents a good trade-off between depth and number of parameters, and is very suitable for parallelization. According to the current results, we established that ResNet-50 is not suitable to deal properly with the information provided by all the multispectral bands. However, a more detailed study (i.e., out of the scope of this work) should be conducted by considering different experimental classification settings (e.g., compare the classification result obtained with one band against RGB).

As has been stated at the introduction of this paper, DL poses challenging questions in terms of time required for the training of a model due to the large number of parameters. The results presented in Section 4.2, confirmed that the Horovod distributed training framework enables the achievement of near linear scaling. However, when dealing with distributed training, the consistency of the classification results has to be constantly monitored. The reason is that when the size of the batch is increased (defined as $b_e = b_g \times k$, where $b_e$ is the effective batch size, $b_g$ is the batch size per GPU, and $k$ is the number of GPUs) a degradation of the accuracy often occurs. At first glance in Figure 6, a slow trend of a decrease in the fscore is apparent when a larger number of nodes is employed. The results obtained using JUWELS are confirmed also by those from JURECA (please note that the fscore of 1 node is not reported, since the computation time has exceeded the limit of the system). Without further special mechanisms, stable training with SGD is possible only for a total batch size of <8192 [66]. During training, an explosion of the loss during the first epochs with a high learning rate was typically observed, which does not occur at a more advanced stage of the training when a lower learning rate is used. This phenomenon is particularly noticeable when a large number of nodes is used. The initial learning rate is in fact dependent on the number of nodes in the formula shown in Section 3.5. As a direct consequence, if a large number of nodes is used, the initial learning rate is large. The step decay learning rate scheduler used in the present work is the one defined by Goyal et al. [50]; however, different schemes such as the polynomial decay scheduler could be employed to make the loss less prone to explosion during the training process. The use of different types of optimizers could as well be studied further in detail as a workaround to overcome this known problem.

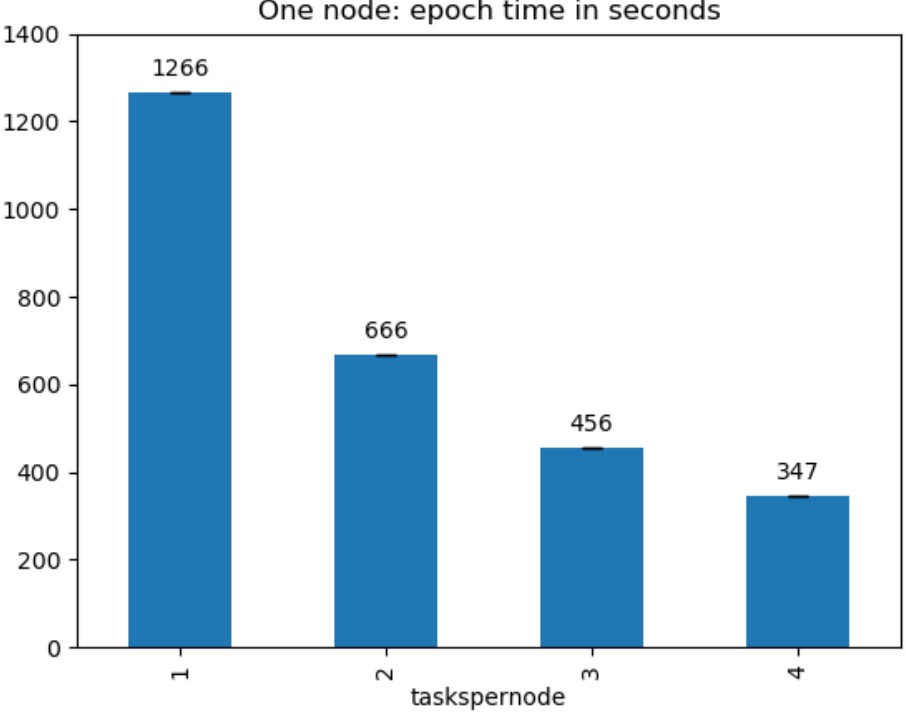

**Figure 3.** JUWELS: One node, 1, 2, 3 and 4 GPUs, time per epoch, multispectral model.

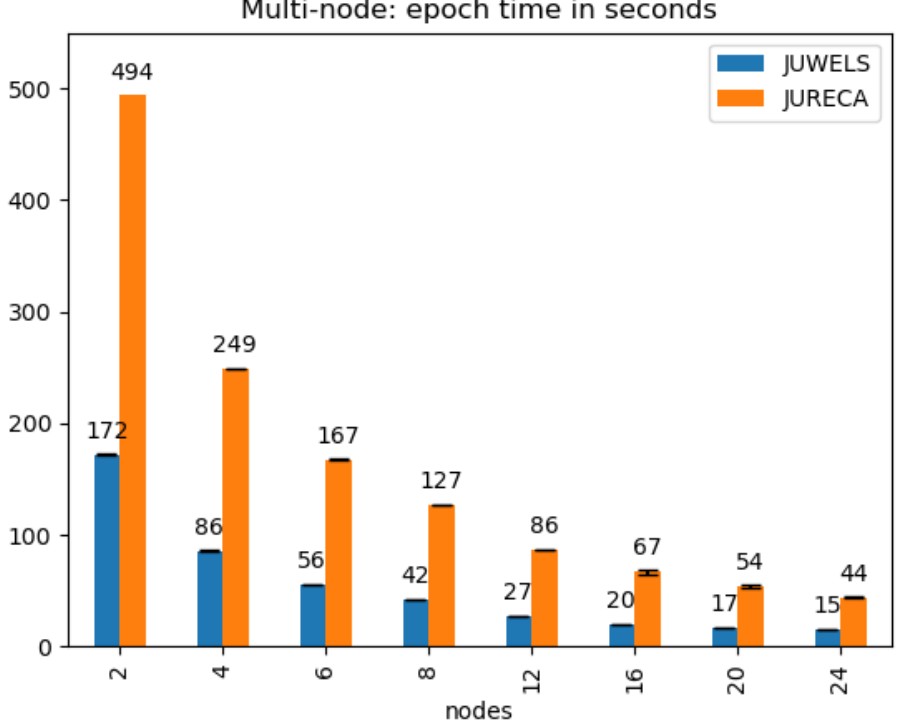

**Figure 4.** Multinode, time per epoch, multispectral model.

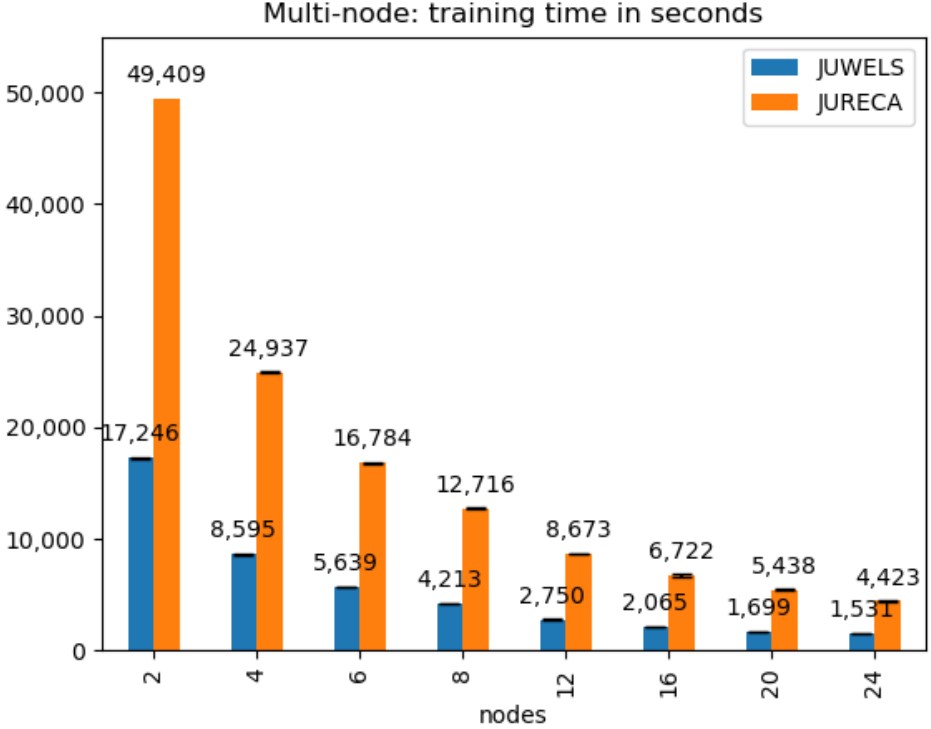

**Figure 5.** Training time, multispectral model.

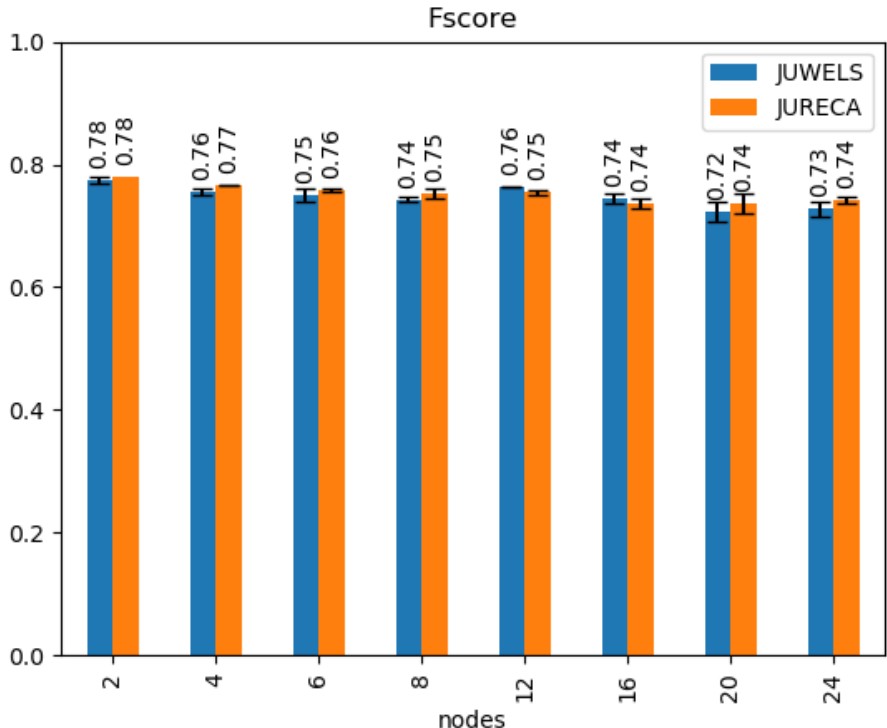

**Figure 6.** Fscore, multispectral model.

## 6. Conclusions

Large-scale deep neural networks have millions of weights and require large amounts of data to optimize these parameters to converge to a satisfactory testing accuracy. With the size of the learning networks and annotated remote sensing datasets growing, it becomes possible to automatically extract useful features and representations suitable for high-accuracy classification tasks, but at the cost of higher computation time necessary for the full training. The experimental results of this paper confirm that distributed training over HPC systems can drastically reduce the amount of time needed to complete the training step, resulting in near linear scaling without significant loss of test accuracy. The publication of this paper includes the availability of the dataset and the Python implementation of the models (https://gitlab.com/rocco.sedona/mdpi-paper-bigearth).

**Author Contributions:** Data curation, R.S. and G.C.; Conceptualization, investigation, formal analysis, methodology, writing—original draft preparation, R.S., G.C., and J.J.; Experiment adjustment, R.S., G.C., J.J., and A.S; Supervision, writing—review and editing, G.C., J.J., and A.S.; Project administration and funding acquisition, M.R. and J.A.B.

**Funding:** This research received no external funding.

**Conflicts of Interest:** The authors declare no conflict of interest.

## Abbreviations

The following abbreviations are used in this manuscript:

| | |
|---|---|
| EO | Earth Observation |
| RS | Remote Sensing |
| DL | Deep Learning |
| ML | Machine Learning |
| HPC | High-Performance Computing |
| MPI | Message Passing Interface |
| CNN | Convolutional Neural Network |
| RNN | Recurrent Neural Network |
| GAN | Generative Adversarial Network |
| MS | Multispectral |
| ResNet | Residual Network |
| JUWELS | Jülich Wizard for European Leadership Science |
| JURECA | Jülich Research on Exascale Cluster Architectures |
| GPU | Graphics Processing Unit |
| CPU | Central Processing Unit |
| SGD | Stochastic Gradient Descent |
| CLS | CORINE Land Cover |

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
