# Peer review of "Remote Sensing Big Data Classification with High Performance Distributed Deep Learning"

_remotesensing, doi:10.3390/rs11243056_

Round 1

Reviewer 1 Report

Suggest to condense the paper and somehow heighten the significance of your experimental results.  As is, the paper is long on details (description of datasets, preprocessing, experiments, results) but short on important or even interesting findings.  "The experimental results confirm that distributed training can drastically reduce the amount of time needed to perform full training, resulting in near linear scaling without loss of test accuracy." sounds hardly significant.   Can you analyze and present the speedup in more quantitative terms?

Reviewer 2 Report

This paper shows that the training of state-of-the-art deep Convolutional Neural Networks (CNNs) can be efficiently performed in distributed fashion using parallel implementation techniques on HPC machines containing large number of Graphics Processing Units (GPUs). The authors need to address the following issues before a possible publication.

How does the author solve the issue regarding efficient communication of information necessary for model updates during the distributed training across multiple nodes? How does the author solve optimization problem during distributed training?

In section 3.2, are there any criteria for determining the number of nodes? In section 3.5, “The input size has been changed from the original size of 224x224 pixels for each image to the size of the patches”. Since the database used by the author is BigEarthNet, the size of the image is resized to 224x224, will there be quality distortion of the image and will it affect the classification performance? Is there any other similar distributed training framework? If such jobs exist, the authors can make a comparison.

Reviewer 3 Report

This manuscript deals with distributed training of deep neural networks for remote sensing image classification. The authors focus on Sentinel-2 image classification using the BigEarthNet dataset. They investigate how to deal with the large volume of data that are produced every month by the S2 constellation and how HPC can deal with those problems. Experiments with ResNet-50 demonstrate that distributed optimization can significantly reduce the training time with a moderate accuracy decrease.

This paper investigates an important problem: can DL scale with the sheer amount of data we face in Earth Observation? Although it is far from solving it, it is a first step in the right direction to achieve scalable deep learning for remote sensing. It is there definitely of interest for readers of the journal.

There are a few English mishaps in the paper that an additional proofreading could catch and figures could be somewhat improved, however the presentation is fairly good.

My main comments are related to the motivation of the experimental setup and potential research directions:
1. Why choose the Hovorod software library? Are there other option for decentralized training? If yes, what makes Hovorod the correct choice in this case?
2. What is the train/val/test split of the BigEarthNet dataset?
3. Most of the manuscript focuses on reducing the training time. Yet, inference time is the bottleneck for real-time processing of EO data. Does Hovorod helps in this regard? By simple duplicating of the network on multiple nodes, one could expect a linear scaling of the inference throughput. Is that the case? This might seem like a naive question but inference is not mentioned in the paper.
4. Since the input image sized is half the one from ImageNet, one could remove the maxpooling layer after the first convolution to preserve the spatial information. Is that the case here? If not, could that significantly change the results?
5. There are six classes for which the F1 score is lower using multispectral data compared to RGB: airports, bare rock, peatbogs, port areas, sea and ocean, sports and leisure activities. Some are man-made objects which have distinct appearances in RGB but similar spectral signatures so the confusion might be understandable. However I am perplexed by the fact that F1 score on sea & ocean is lower using multispectral input than using RGB. Could the authors comment on these classes?
6. The authors mention that there is a significant class unbalance. Data augmentation will not really help in this regard since it is applied equally on all classes. However a simple loss reweighting can significantly improve the F1 score on the less represented classes. I would be interested by an experiment in this regard (at least to see if it makes a difference or if more sophiscated approaches are indeed necessary).

I also have a few minor comments:
a. Does the super-resolved images help achieve a higher accuracy compared bilinear interpolation upsampling?
b. Are the mean and standard deviation computed on 2 runs significant?
c. Figures 3/4, 5/6, etc. could be improved by reporting the actual values over the bars and by grouping the plots that belong to the same experiment.
d. In the introduction (esp. in Table 1), it should be made clear that this work addresses image classification and not semantic mapping (which is often the ultimate goal in EO).

And finally I have some nitpicks:
* l.36: "this data do not exist" -> "these data do not" or "this data does not"
* l.42: "raw hyperspectral" -> "raw multispectral"?
* l.54: ImageNet contains many images but the reference ILSVRC challenge uses 1 000 000 images (1000 classes with 1000 samples).
* l.74: "enable to speed-up" -> "speeds-up"?
* l.78: "this enables the possibility of deploying" -> "this enables deployment of"
* l.152: "were gradients" -> "where gradients"
* l.165: "note failure" -> "node failure"
* Numbers such as 590,326 could be easier to parse with a comma.
* l.224: "scikit 0.20.3" -> I assume this is scikit-learn?
* l.320-321: "also the class..., still is easily" -> "the class"..., is also easily

Round 2

Reviewer 2 Report

Previous comments have been addressed and I recommend to accept this paper.